# The Effect of Competency-Based Triage Education Application on Emergency Nurses’ Triage Competency and Performance

**DOI:** 10.3390/healthcare10040596

**Published:** 2022-03-22

**Authors:** Sun-Hee Moon, In-Young Cho

**Affiliations:** College of Nursing, Chonnam National University, 160 Baekseo-ro, Dong-gu, Gwang-ju 61469, Korea; sunnymon@jnu.ac.kr

**Keywords:** triage, mobile applications, education, distance, competency-based education, KTAS

## Abstract

The Korean Triage and Acuity Scale (KTAS) is used to determine emergency patient priority. The purpose of this study was to develop the Competency-Based Triage Education Application (CTEA) using KTAS and evaluate its effectiveness on emergency nurses’ triage competency and performance. The developed CTEA mobile application comprised 4 lectures, 12 text-based cases, and 8 video-based triage scenarios. A quasi-experimental pre-post design with a comparison group (CG) was used to evaluate the effectiveness of the CTEA. Thirty-one participants were assigned to an intervention group (IG) and used the application for at least 100 min over one week. Thirty-five participants were assigned to a CG and underwent book-based learning, which covered the same content as the CTEA. Triage competency (t = 2.55, *p* = 0.013) and performance (t = 2.11, *p* = 0.039) were significantly improved in the IG. The IG’s undertriage error was significantly reduced compared to that of the CG (t = 2.08, *p* = 0.041). These results indicated that the CTEA was effective in improving the emergency nurses’ triage competency and performance. This application will be useful as a program for providing repeated and continuous triage education.

## 1. Introduction

In modern emergency departments (EDs), the triage process is applied to quickly identify patients’ acuity and classify their priority based on the severity of their conditions [1,2]. In EDs, emergency treatment is provided based on urgency rather than on a first-come, first-served basis; therefore, triage, which guides the allocation of limited medical resources—including health care providers and available medical equipment—according to patient’s conditions, is an essential system for patient safety [1,2]. Typically, nurses are the main practitioners of triage in EDs, acting as gatekeepers at ED entrances [2,3]. Hence, educational support to ensure sufficient triage competency among emergency nurses is imperative.

Due to its proliferation, triage scales have been developed; in particular, a five-tier triage scale that divides patient urgency into five levels is mainly applied in modern EDs [4]. The triage scales used worldwide include the Canadian Triage and Acuity Scale (CTAS), the U.S. Emergency Severity Index (ESI), the Australasian Triage Scale (ATS), and the U.K. Manchester Triage System (MTS) [4]. In addition to these, a triage scale suitable for each country’s emergency medical system has recently been developed. In 2016, the Korean Triage and Acuity Scale (KTAS) was established based on CTAS and used in EDs nationwide [5]. To ensure that nurses can use the KTAS to make accurate and rapid decisions, appropriate educational support is required.

Various training methods have been applied to assist triage nurses in making accurate and rapid decisions. According to existing review studies on triage education, the methods used primarily include brief lectures with case studies, simulations, live actors, computerized scenarios, and game-based learning [6,7,8]. Since the goal of triage education is to improve emergency nurses’ competence in making accurate and rapid decisions through various educational methods [2,7,9], there is a need for more efficient and learner-friendly methods to be developed. Countries that have created a representative triage scale offer efficient triage education online [6]; recently, various techniques, such as virtual reality and augmented reality, have been introduced [10,11].

A few KTAS-based education programs have been developed, including the official hospital-KTAS provider course, an offline training program consisting of lectures and text-based case studies [12]. However, one-time education programs offered offline suffer from certain limitations. Specifically, to improve their triage competencies, emergency nurses need to experience triage repeatedly and continuously [3], which one-time courses do not offer. According to extant review studies, smartphone-based education significantly improved nursing and medical science practitioners’ clinical competency, knowledge, attitude, performance, and confidence [13,14]. Since online education via mobile devices can be accessed freely and repeatedly without spatial and temporal constraints, a positive effect can be expected for app-based triage education centered on the KTAS.

Therefore, the purpose of this study was to develop a mobile-app-based online education program to improve emergency nurses’ triage competency and verify the effectiveness of the education program. This study hypothesized that participants using the Competency-Based Triage Education Application (CTEA) would have greater (1) triage competency and (2) triage performance, compared to those learning triage using a book.

## 2. Materials and Methods

### 2.1. Design

This was a pre- and post-test quasi-experimental study with a comparison group, designed to evaluate the effect of the CTEA on emergency nurses’ triage competency and performance.

### 2.2. The CTEA Development

The basic framework of the CTEA was designed based on a literature review and a qualitative study conducted by the research team to explore the educational needs of emergency nurses. The development of the preliminary educational contents was led by the first author, who has 12 years of experience working in EDs and more than 6 years of triage experience. The lecture content was developed with reference to four triage books from the Emergency Nurses Association and KTAS committees and one emergency medicine book [15,16,17,18,19]. The educational content included four lectures, 12 text-based case studies (covering abdominal pain, dyspnea, fever, vomiting, epistaxis, simple treatment, cough, behavior, dizziness, vaginal bleeding, and traffic accidents), and eight conversational scenarios (covering acute myocardial infarction, acute stroke, major trauma, congestive heart failure, acute hemorrhage, urticarial rash, appendicitis, and hyperventilation) (Figure 1). The case study was designed to equip the learners to solve the competency targeted KTAS quizzes. The CTEA was structured such that learners could ask the operator questions about triage cases and communicate freely with each other (learner to learner).

An expert group consisting of two emergency medicine professors and four emergency nursing professors who evaluated the validity of the educational content using content validity indices (CVIs). The overall CVI score of the educational contents was 0.92. In accordance with the experts’ opinions, the vital signs of the triage cases and the dialogue of the scenario were modified.

The educational content was then converted to video form. Specifically, the first author created a 50 min video containing four lectures. The actors filmed the conversational scenarios in 8 videos, each 2–3 min long. The developed educational content was loaded into an Android-based hybrid mobile app (Figure 2). Learners were able to score “energy” points when viewing triage lectures, taking quizzes, and reviewing cases using the app, and they were able to check their ranking against other learners.

### 2.3. Outcome Measurements

To determine the nurses’ KTAS competency, we measured four outcomes—critical thinking disposition, triage competency, triage knowledge, and triage performance.

Critical thinking disposition was the propensity to think critically to lead decision-making and problem-solving tasks [20]. In this study, critical thinking disposition was measured using a 5-point Likert scale developed by Kwon et al. in 2006 [20]. The scale consists of 35 items with potential total scores ranging from 35 to 175 points, with higher scores indicating higher critical thinking disposition. At the time of development, the scale received a Cronbach’s α of 0.89 [20]. In this study, it received a Cronbach’s α of 0.83.

Triage competency is the ability to allocate medical resources efficiently by determining care priority according to patients’ health statuses [1]. In this study, triage competency was measured using a 5-point Likert scale developed by Moon et al. in 2018 [21]. The triage competency measure comprises 30 items with five sub-factors, including clinical judgment, expert assessment, management of medical resources, timely decisions, and communication [21]. The potential scores of the scale range from zero to 150, with higher scores indicating higher triage competency. The Cronbach’s α of the scale in Moon et al.’s study was 0.91 [21], while in this study it was 0.96.

To measure triage knowledge, we developed a scale consisting of 32 preliminary items based on the four triage books mentioned previously. The content validity verification of the developed items was conducted by a panel of experts who participated in the CTEA’s content validity verification. Among the 32 preliminary items, one item with a CVI score of 0.67 was deleted, resulting in 31 final items; the CVI of the final triage knowledge scale was 0.96. The items were answered by selecting one true/false answer for each of the 31 questions, and triage knowledge was calculated by totaling up the correct answers. The KR-20 of the triage knowledge scale was 0.56.

To measure triage performance we created 10 triage scenarios, drafted based on the experience of the first author and revised through research meetings. An expert panel verified the content validity of the scenarios written in conversational format. The CVI of the triage scenarios was 0.93. Videos, which were approximately 2 min in length, were created for the 10 triage scenarios, to which participants had to respond by assigning the videos a KTAS-based score from level 1 to level 5. The number of correct answers would then be summed up, overtriage error totaled up, and undertriage error totaled up. The KR-20 of the triage performance scale was 0.71.

### 2.4. Participants and Data Collection

Six EDs in four cities participated in this study from July 2020 to January 2021. The emergency medical system in Korea consists of the regional emergency center (REC), which is in charge of treating critical emergency patients, and the local emergency center (LEC), which is in charge of treating emergency patients in their local area. Of the EDs participating in this study, two were RECs and four were LECs. All EDs have been performing triage using KTAS since 2016. Three EDs each were assigned to the intervention and the comparison group (Figure 3). Thirty-eight emergency nurses were assigned to the intervention group and 37 to the comparison group. In total, 75 emergency nurses in the EDs responded to our online survey comprising six measures—critical thinking, triage competency, triage knowledge, triage performance, demographic factor (age, gender, educational level), and clinical experience (total nursing, emergency nursing, triage) before intervention.

The participants in the intervention group were provided with instructions on installing the CTEA. The participants read the guide and then downloaded and installed the CTEA from Play Store. Android smartphones were rented for Apple smartphone users. The participants in the intervention group underwent triage training using the CTEA for over 100 min in one week. The lengths of time the intervention group participants spent learning via the CTEA were saved in the program so that the administrator could monitor them. Five of the learners who studied for less than 100 min over the week were excluded from the intervention group. After one week of using the CTEA, the intervention was deemed complete and a post-survey in the same format as the pre-survey was conducted online. Of the 33 participants who completed learning in the intervention group, two participants did not respond to the post-survey and were excluded. The remaining 31 participants’ data were analyzed.

For the 37 participants in the comparison group, the 4 lecture materials and 12 text-based case studies (containing the same content as the CTEA material) were provided as a book. The comparison group participants completed 100 min of self-learning over one week using the triage book. The participants measured and reported their learning time to the researchers. After completing the learning, a post-survey in the same format as the pre-survey was conducted. Two of the comparison group participants were excluded due to insufficient self-learning time. The remaining 35 participants’ data were analyzed.

The sample size calculation yielded G*Power 3.1.9.4. According to a meta-analysis on virtual patient education for health professionals, the effect size *d* was 0.8 [22]. In a two-tailed significance test with a power of 80% and an alpha level of 0.05, the sample size of each group was calculated to be 26. Therefore, the data used in this study’s analysis met this criterion.

### 2.5. Data Analysis

The data were analyzed using IBM SPSS software, version 25.0(Armonk, NY: IBM Corp). The variables were analyzed based on frequencies, percentages, or means. We compared the baseline data of the two groups using chi tests for the categorical variables and *t*-tests for the continuous variables. To test the differences in critical thinking disposition, triage competency, triage knowledge, and triage performance, we calculated the difference between the pre-post means within each group and then compared the differences between the two groups using the independent *t*-test. As the pre-test result presented a difference in baseline triage knowledge, ANCOVA was applied to verify the difference in post-knowledge between the two groups using pre-knowledge as a covariate.

### 2.6. Ethical Approval

This study received approval from the University Institutional Review Board (No. 7001066-202002-HR-006). All participants willingly engaged in the data collection and intervention process and signed online informed consent forms. As a token of appreciation, all participants were awarded a $50 gift card at the end of the study.

## 3. Results

### 3.1. Baseline Characteristics

The average age of the participants was 32.15 ± 7.47 years, and most of them (56) were women (Table 1). The average total nursing experience was 8.85 ± 7.54 years. In accordance with the Career Development Model of Nurses, a clinical career was defined in four-stages: novice (under 1 year of nursing experience), advanced beginner (1~3 years of nursing experience), competent (3~7 years of nursing experience), and proficient nurses (7 or more years of nursing experience) [23]. The participants’ average emergency experience was 4.04 ± 3.54 years, with 24 (36.4%) of them falling within the competent level. The participants’ triage experience was 1.25 ± 1.46 years. Except for the triage knowledge score, the baseline variables of the two groups before intervention did not differ (t = 2.56, *p* = 0.013) (Table 2).

### 3.2. Comparison of Outcomes

The first hypothesis in this study posited that the intervention group participants (those using the CTEA) would show higher triage competency than those in the comparison group. The intervention group’s pre-post triage competency levels showed significant improvement compared with that of the comparison group (t = 2.55, *p* = 0.013) (Table 3). Specifically, after examining the sub-factors of the pre-post triage competency, the intervention group participants’ clinical judgment (t = 2.39, *p* = 0.021) and timely decisions (t = 2.89, *p* = 0.005) showed significant improvement.

The second hypothesis of this study was that the intervention group participants would show higher triage accuracy than those in the comparison group. When the pre-post triage accuracy of the two groups was verified, the intervention group showed significant improvement (t = 2.11, *p* = 0.039). To evaluate the cause of triage error, the averages of the two groups’ undertriage and overtriage were investigated and compared (Figure 4). The results revealed a significant reduction in the intervention group’s undertriage error, compared to the comparison group (t = 2.08, *p* = 0.041). However, both groups showed a decrease in the average of overtriage error.

There was no significant difference between the two groups regarding critical thinking disposition when the pre- and post-test values were compared (t = −0.47, *p* = 0.633). For triage knowledge, after controlling the baseline values, that is, the covariate, no difference was found (F = 3.52, *p* = 0.065).

## 4. Discussion

Due to the spread of COVID-19, various non face-to-face education methods have recently been developed and utilized. Smartphone-based learning is one of the representative non face-to-face education methods that emphasizes learner accessibility. Recent research has reported that smartphone-based education improves clinical competency, knowledge, performance, attitude, and confidence in nursing and medical science practice [13,14]. Since a smartphone-based triage education program was developed, distance training and repeat clinical decision-making practice was rendered possible [24,25]. The significance of this study was the development of a mobile education app for strengthening the triage competency of emergency nurses based on the KTAS and the verification of its effectiveness. This KTAS-based educational mobile app development and effectiveness verification was the first attempt in Korea. Furthermore, since the importance of repetitive education is emphasized for the advancement of triage [3], the CTEA can be a complementary method to official triage education that allows iterative and convenient learning.

The Emergency Nurses Association (ENA) establishes practice guidelines for triage qualifications; thus, it is necessary to strengthen triage competency by improving knowledge and skills through appropriate triage education courses [26]. Extant research also reported competency and knowledge improvement as the main effect of triage education. Previous research reported that nursing students’ competency and knowledge significantly improved after they were provided with education, including role plays and lectures based on the Simple Triage and Rapid Treatment (START) triage [27]. In a disaster nursing education program comprising lectures and practical tasks, nursing students’ disaster nursing knowledge, disaster triage performance, and disaster readiness increased [28]. In this study, after undergoing CTEA-based training, the participants’ triage competency improved. However, although the mean difference in triage knowledge between the two groups in this study was significant, there was no significant difference when the baseline score was adjusted using ANCOVA. Compared to the previous face-to-face research, the CTEA was conducted in a non face-to-face manner, which, except for observing learning time, translated to the limited monitoring of learners. Thus, triage knowledge may not have improved. Additionally, the intervention group’s triage knowledge pre-test score and the comparison group’s triage knowledge pre- and post-test score were almost similar. This may be due to the ceiling effect.

Triage competency comprises five attributes: clinical judgment, expert assessment, management of medical resources, timely decisions, and communication [1]. Among the sub-factors of triage competency, clinical judgment and timely decisions showed significant improvement in this study. When emergency nurses perform triage, rapid and accurate decision-making is important, making it a crucial objective that must be achieved through triage education [1,7,18,26]. Therefore, improvements in clinical judgment and the ability to make timely decisions through the CTEA could be referred to as educational goals. However, sub-factors such as communication and expert assessment did not significantly improve in this study. As physical examination requires direct practice, it may have been difficult to achieve the same in mobile-based non face-to-face research. Other than the chat function, the CTEA did not have a feature that allowed participants to communicate with patients or medical staff. Due to these development limitations, communication ability may not have been improved. In cases where artificial intelligence plays the role of an emergency patient or where learners perform triage simulations with standard patients in a virtual space, improvements in learners’ communication in non face-to-face education could be expected.

According to a review study, triage accuracy was reported to be 56.2~82.9% when analyzed based on written case scenarios or medical record review results [29]. Based on electronic medical records in the EDs using KTAS, the weighted kappa value was 0.69~0.83 for triage accuracy, which increased to 0.84 after problem-based learning [30,31]. In this study, triage accuracy significantly increased after using the CTEA; however, compared to a previous study [29], this was not high. Although the mobile-based education in this study had a limited effect on triage accuracy compared to face-to-face education, such education may be a good alternative; as triage education must be accessible and administered continuously and repeatedly to maintain triage accuracy [3] and mobile devices allow for such conveniences. Thus, if various mobile-based triage education programs are developed, triage quality can eventually be achieved and maintained.

There are two types of triage errors: undertriage and overtriage [32,33]. Undertriage is when treatment time for emergency patients is delayed due to underestimating the severity of the patients’ condition at triage, potentially jeopardizing patients’ safety [33]. Overtriage is when patients’ conditions are overestimated, which can cause overcrowding in the ED and jeopardize patients’ safety by endangering urgent patients [32]. Mistriage in the Korean EDs using KTAS resulted in an undertriage of 70.4% and overtriage of 29.6%, which was reported to have a higher underestimation error [31]. In this study, the incidence of undertriage was higher than that of overtriage when assessed using the video-based triage scenarios. The nurses’ triage errors were reported to decrease when the web-based video education program developed using the KTAS was applied [34]. Similarly, in this study, when the mobile-based video scenario was applied, undertriage decreased significantly, and overtriage showed a decreasing pattern, although it was not statistically significant. Therefore, it could be said that the use of video-based scenarios for triage learning was effective and that it is necessary to develop various triage cases to ensure that learners can repeatedly access education more conveniently. In addition, since the incidence of undertriage in Korea’s EDs using the KTAS was high, it may be necessary to develop an education program focused on reducing underestimation.

Gamification is the application of a game design in non-game contexts and has been widely used in education programs to motivate users, enhance psychological outcomes, and encourage behavioral change [35,36,37]. As in many other educational programs, gamification has also been introduced to triage education. A study on trauma triage education using serious game technology reported improvement in emergency physicians’ decision heuristics [38]. According to review studies, game features commonly used in healthcare gamification include points, social interactions, leaderboards, progress statuses, levels, and immediate feedback [36]. The gamification strategies used in this study were “energy points,” social interaction through chatting, levels, and immediate feedback. In a meta-analysis study verifying the effect of gamification in medical education, knowledge improvement and long-term knowledge retention were reported [39]. Although it was not possible to directly verify the educational effect of the gamification in this study design, if the same gamification strategy is used in future triage education, we expect continuity in the training effect.

This study had a few limitations. First, it measured the effectiveness of the CTEA program immediately after its use. Therefore, it was not possible to estimate the continuation of the educational effect on the participants and the interval requiring repeated education. If the effect of continuous education through the CTEA is confirmed, the application can be used more effectively as a complementary program for the official KTAS education. Second, this was a quasi-experimental study and participants were not randomly assigned. Randomization and blinding can assert strong causality and are suitable in research settings such as laboratories [40]. Many randomized studies examining the effectiveness of medical education have raised validity and reliability concerns; therefore, it has been suggested that they be categorized as quasi-experimental studies [40]. In this study, in accordance with the suggestions of a review [40], a randomized controlled trial, which was considered difficult in the educational field, was not unreasonably followed. Although random assignment could not be performed in this study, we attempted to maintain validity by assigning groups based on hospitals to avoid influence between participants and conducted a homogeneity test. Third, the book provided to the CG included 4 lectures and 12 text-based cases, except for the 8 videos provided to the IG. Although video-based contents could not be included in a paper-based book, differences in the amount of content provided may have affected the results. Fourth, this study was conducted only in six EDs in Korea; therefore, there was a limit to the generalizability of the results.

## 5. Conclusions

The development of the CTEA and its application to emergency nurses in this study improved triage competency and performance in the intervention group. Moreover, the CTEA was effective in reducing triage error. Therefore, the CTEA can be used as an educational program for continuous triage education because it is mobile-based and enables repetitive and convenient learning.

## Figures and Tables

**Figure 1 healthcare-10-00596-f001:**
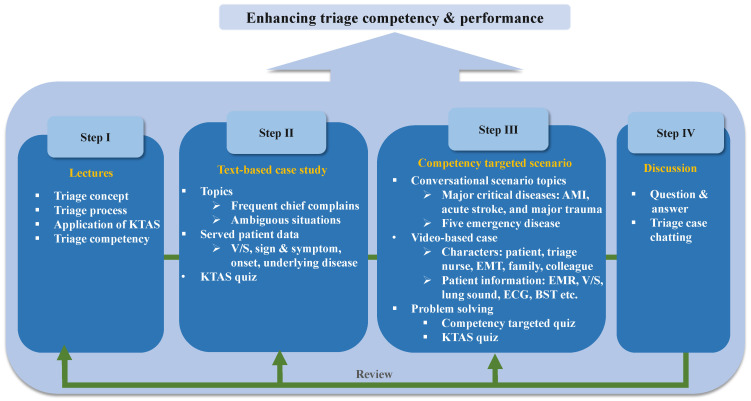
Triage education program. KTAS = Korean Triage and Acuity Scale, V/S = vital sign, AMI = acute myocardial infarction, EMT = emergency medical technician, EMR = electronic medical records, ECG = electrocardiogram, BST = blood sugar test.

**Figure 2 healthcare-10-00596-f002:**
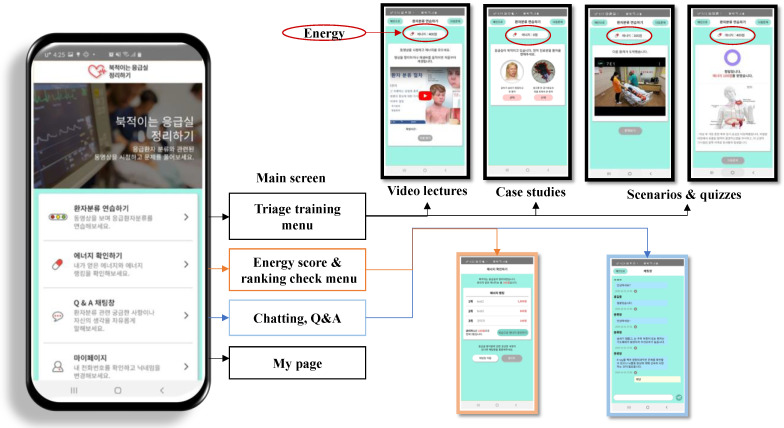
The Competency-Based Triage Education Application.

**Figure 3 healthcare-10-00596-f003:**
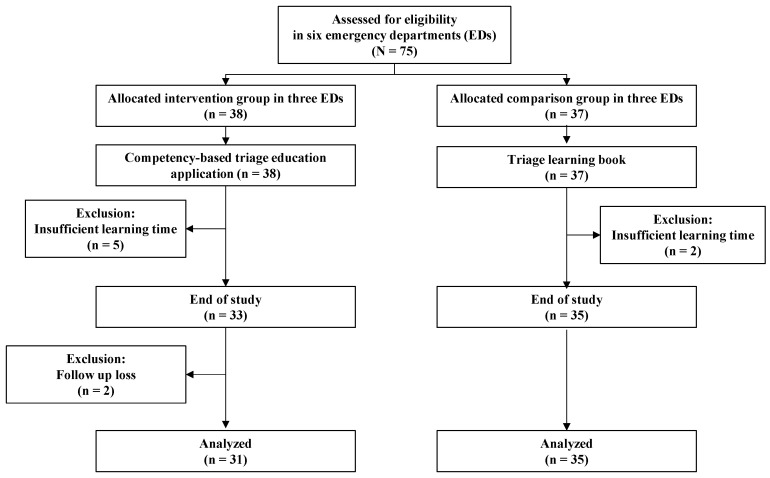
Participant’s flow chart.

**Figure 4 healthcare-10-00596-f004:**
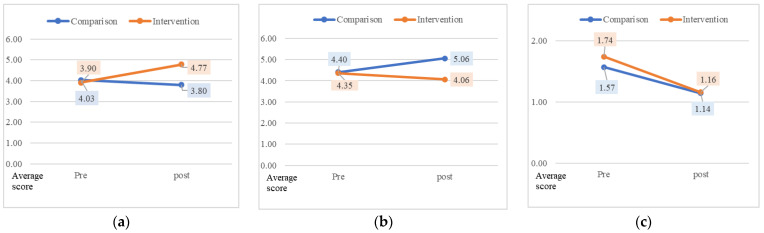
Triage accuracy and causes of triage error. (**a**) Triage accuracy. (**b**) Undertriage. (**c**) Overtriage.

**Table 1 healthcare-10-00596-t001:** Baseline characteristics of participants.

Characteristics	Classification	IG (n = 31) n (%) or Mean ± SD	CG (n = 35) n (%) or Mean ± SD	Total (N = 66) n (%) or Mean ± SD	*χ*2 or t (*p*)
Age	Total (years)	34.09 ± 8.19	30.42 ± 6.41	32.15 ± 7.47	2.00 (0.050)
	20~29	11 (16.7)	20 (30.3)	31 (47.0)	4.66 (0.097)
	30~39	12 (18.2)	12 (18.2)	24 (36.4)	
	≥40	8 (12.1)	3 (4.5)	11 (16.7)	
Gender	Female	26 (39.4)	30 (45.5)	56 (84.8)	0.04 (0.834)
	Male	5 (7.6)	5 (7.6)	10 (15.2)	
Education level	Associate degree	11 (16.7)	5 (7.6)	16 (24.2)	4.93 (0.085)
	Bachelor’s degree	18 (27.3)	29 (43.9)	47 (71.2)	
	Over master’s degree	2 (3.0)	1 (1.5)	3 (4.5)	
Experience in nursing	Total (year)	10.46 ± 8.19	7.42 ± 6.72	8.85 ± 7.54	1.65 (0.104)
	Novice (<1)	2 (3.0)	2 (3.0)	4 (6.1)	2.53 (0.471)
	Advanced beginner (1≤~<3)	4 (6.1)	7 (10.6)	11 (16.7)	
	Competent (3≤~<7)	6 (9.1)	11 (16.7)	17 (25.8)	
	Proficient (≥7)	19 (28.8)	15 (22.7)	34 (51.5)	
Experience in the ED	Total (year)	4.06 ± 3.12	4.02 ± 3.93	4.04 ± 3.54	0.05 (0.960)
	Novice (<1)	5 (7.6)	5 (7.6)	10 (15.2)	0.13 (0.989)
	Advanced beginner (1≤~<3)	9 (13.6)	11 (16.7)	20 (30.3)	
	Competent (3≤~<7)	11 (16.7)	13 (19.7)	24 (36.4)	
	Proficient (≥7)	6 (9.1)	6 (9.1)	12 (18.2)	
Experience of triage (year)		1.27 ± 1.77	1.23 ± 1.14	1.25 ± 1.46	0.10 (0.920)

IG = intervention group, CG = comparison group, SD = standard deviation, ED = emergency department.

**Table 2 healthcare-10-00596-t002:** Baseline outcome variables of participants.

Variables	IG (n = 31) Mean ± SD	CG (n = 35) Mean ± SD	Total (N = 66) Mean ± SD	t *(p)*
Critical thinking disposition	113.70 ± 9.62	112.34 ± 10.28	112.34 ± 10.28	−0.55 (0.581)
Triage competency	79.51 ± 16.68	82.88 ± 12.63	82.88 ± 12.63	0.93 (0.355)
Triage knowledge	20.35 ± 3.15	22.22 ± 2.77	22.22 ± 2.77	2.56 (0.013 *)
Triage accuracy	3.90 ± 1.86	4.02 ± 1.85	4.02 ± 1.85	0.27 (0.786)

IG = intervention group, CG = comparison group, SD = standard deviation, ED = emergency department, * *p* < 0.05.

**Table 3 healthcare-10-00596-t003:** Comparison between pre- and post-outcomes of the two groups.

Variables	Groups	Pre (a) Mean ± SD	Post (b) Mean ± SD	Difference (b − a) Mean ± SD	t (*p*)
Critical thinking disposition	IG	113.70 ± 9.62	115.45 ± 10.31	1.74 ± 8.89	−0.47 (0.633)
	CG	112.34 ± 10.28	115.14 ± 10.93	2.80 ± 8.98	
Triage competency	IG	79.51 ± 16.68	86.25 ± 15.79	6.74 ± 14.42	2.55 (0.013 *)
	CG	82.88 ± 12.63	82.54 ± 12.65	-0.34 ± 7.40	
Clinical judgment	IG	34.12 ± 6.65	37.00 ± 6.37	2.87 ± 6.41	2.39 (0.021 *)
	CG	35.62 ± 5.33	35.45 ± 5.26	−0.17 ± 3.13	
Expert assessment	IG	10.00 ± 3.01	11.09 ± 2.59	1.09 ± 2.97	1.22 (0.225)
	CG	9.85 ± 2.46	10.20 ± 2.13	0.34 ± 1.79	
Management of medical resources	IG	11.29 ± 2.84	11.87 ± 2.26	0.58 ± 2.87	1.99 (0.051)
	CG	12.22 ± 2.27	11.62 ± 2.34	−0.60 ± 1.89	
Timely decisions	IG	10.51 ± 2.95	11.41 ± 2.93	0.90 ± 1.90	2.89 (0.005 *)
	CG	10.71 ± 2.29	10.31 ± 2.45	−0.40 ± 1.75	
Communication	IG	13.58 ± 3.26	14.87 ± 3.09	1.29 ± 3.01	1.30 (0.197)
	CG	14.45 ± 2.44	14.94 ± 2.02	0.48 ± 1.72	
Triage knowledge	IG	20.35 ± 3.15	22.41 ± 2.72	2.06 ± 2.95	3.11 (0.003 *)
	CG	22.22 ± 2.77	22.28 ± 2.39	0.05 ± 2.27	
Triage accuracy	IG	3.90 ± 1.86	4.77 ± 1.68	0.87 ± 2.34	2.11 (0.039 *)
	CG	4.03 ± 1.85	3.80 ± 1.71	−0.22 ± 1.88	

IG = intervention group, CG = comparison group, SD = standard deviation, * *p* < 0.05.

## Data Availability

Not applicable.

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
