# Peer review of "The Effect of Competency-Based Triage Education Application on Emergency Nurses’ Triage Competency and Performance"

_healthcare, 2022, doi:10.3390/healthcare10040596_

Round 1
Reviewer 1 Report
The authors have presented a quasi-experimental study with a comparison group, designed to evaluate the effect of a Competency-Based Triage Education Application (CTEA) on emergency nurses’ triage competency and performance. This study hypothesized that participants using the CTEA would have greater triage competency and triage performance, compared to those learning triage using a book.
Below are suggestions/comments to improve the manuscript.
General concepts comments:
- The greatest disadvantage of this quasi-experimental study is that randomization was not used. Even if the authors have mentioned this as a limitation of the study, I would suggest the authors to be cautious in concluding a causal association between an intervention (development of the CTEA and its application to emergency nurses) and an outcome (triage competency and triage performance).
Specific comments:
- Abstract, Page 1, Line 7 - I would suggest the authors to rephrase the sentence “to classify the urgency of emergency”.
- Page 3, Line 86 – There appears to be a mistake “emergency nurses”.
- Page 5, Line 175 – I would suggest the authors to extend the “Data analysis” section as appropriate.
- Table 1 – I would suggest the authors to separate baseline characteristics and outcomes variables into 2 different tables.
- Table 1 – Please add “n”(%) along with mean±SD.
Author Response
The co-authors and I sincerely appreciate and respect the reviewers’ opinions. Therefore, we have revised our manuscript as best as we could to reflect the reviewers’ suggestions. The edits made are marked in yellow in the manuscript.

Reviewer 2 Report
There is something fundamentally wrong with the paper based on Table 2. Many of the pre-post differences are negative, suggesting negative learning. Many are also not statistically significant -- I didn't check all of them. Yet, the authors go on to check the statistical significance of the difference between IG and CG and subsequently draw inferences based on the same. There must be a better explanation of this if one must accept it.
Author Response

(The authors gave the same response as above.)

Reviewer 3 Report
The article is very well written, and the English language is superb.
I have only minor comments that can be addressed at proof-reading stage:
I would recommend to explicitly mention “mobile application” in the abstract.
On page 3, line 86, there is an artifact “emergency nurses”.
Figure 1 was inserted with the distortion of the font. Please check.
On page 4, line 147, there is an artifact “intervention-”: “-“.
Figure 3: extra spaces before EDs and no space before (EDs).
If possible, consider additionally addressing in the Discussion:
Based on the app description, the authors used gamification components – energy points and scoring against other learners. Can you elaborate how it might help with attractiveness of the learning. Overall, opine on the attractiveness of your product, if there was any feedback from the users. It is a great study, I would love it to become not just research, but something used widely, so it is important to think about the implementation and scaling and what would be contributing to that. Especially characteristics of the developed app. The effectiveness of training is very well demonstrated but probably this article also can be used to persuade others to adopt the app by boasting about other app features, for example, it is fun and game-like learning.
“Android smartphones were rented for Apple smartphone users.” Is it possible to conduct a sub-analysis of those who got rented smartphones if they did better or worse than their peers? It seems to be an additional incentive, but also more learning to use a completely different phone. Or just report in one line that there was no difference observed. The authors might consider using this fact to additionally ‘boast” how easy it is to use this app even using a completely different phone.
However, if the authors use other means to disseminate about this app as a commercial product and this manuscript aims to solely present evaluation results, please disregard these my comments.
Author Response

(The authors gave the same response as above.)

Round 2
Reviewer 2 Report
The authors do not address my criticism. Many of the pre-post differences are negative, suggesting negative learning. Many are also not statistically significant -- I didn't check all of them. Yet, the authors go on to check the statistical significance of the difference between IG and CG and subsequently draw inferences based on the same. There must be a better explanation is not adequate. I cannot accept it.
Author Response
We added a detailed response to the reviewer's comments.
